# The Impact of Adenomyosis on Pregnancy

**DOI:** 10.3390/biomedicines12081925

**Published:** 2024-08-22

**Authors:** Panagiotis Tsikouras, Nektaria Kritsotaki, Konstantinos Nikolettos, Sonia Kotanidou, Efthymios Oikonomou, Anastasia Bothou, Sotiris Andreou, Theopi Nalmpanti, Kyriaki Chalkia, Vlasios Spanakis, Nikolaos Tsikouras, Melda Chalil, Nikolaos Machairiotis, George Iatrakis, Nikolaos Nikolettos

**Affiliations:** 1Department of Obstetrics and Gynecology, Democritus University of Thrace, 68100 Alexandroupolis, Greece; nektaria.97@hotmail.com (N.K.); k.nikolettos@yahoo.gr (K.N.); kotanidou.so@gmail.com (S.K.); eftoikonomou@outlook.com (E.O.); soterisand@hotmail.com (S.A.); theonalmpanti@hotmail.com (T.N.); kikichalkia@yahoo.gr (K.C.); spanakvls@outlook.com.gr (V.S.); nick.tsik.2001@gmail.com (N.T.); x.melda25@gmail.com (M.C.); nnikolet@med.duth.gr (N.N.); 2Department of Midwifery, School of Health Sciences, University of West Attica (UNIWA), 12243 Athens, Greece; natashabothou@windowslive.com (A.B.); giatrakis@uniwa.gr (G.I.); 3Third Department of Obstetrics and Gynecology, Medical School, National and Kapodistrian University of Athens, Attikon Hospital, Rimini 1, 12462 Athens, Greece; nikolaosmachairiotis@gmail.com

**Keywords:** adenomyosis, pregnancy, complications

## Abstract

Adenomyosis is characterized by ectopic proliferation of endometrial tissue within the myometrium. Histologically, this condition is marked by the presence of islands of benign endometrial glands surrounded by stromal cells. The myometrium appears thinner, and cross-sectional analysis often reveals signs of recent or chronic hemorrhage. The ectopic endometrial tissue may respond to ovarian hormonal stimulation, exhibiting proliferative or secretory changes during the menstrual cycle, potentially leading to bleeding, uterine swelling, and pain. Adenomyosis can appear as either a diffuse or focal condition. It is crucial to understand that adenomyosis involves the infiltration of the endometrium into the myometrium, rather than its displacement. The surgical management of adenomyosis is contingent upon its anatomical extent. The high incidence of the disease and the myths that develop around it increase the need to study its characteristics and its association with pregnancy and potential obstetric complications. These complications often require quick decisions, appropriate diagnosis, and proper counseling. Therefore, knowing the possible risks associated with adenomyosis is key to decision making. Pregnancy has a positive effect on adenomyosis and its painful symptoms. This improvement is not only due to the inhibition of ovulation, which inhibits the bleeding of adenomyotic tissue, but also to the metabolic, hormonal, immunological, and angiogenic changes associated with pregnancy. Adenomyosis affects pregnancy through disturbances of the endocrine system and the body’s immune response at both local and systemic levels. It leads to bleeding from the adenomyotic tissue, molecular and functional abnormalities of the ectopic endometrium, abnormal placentation, and destruction of the adenomyotic tissue due to changes in the hormonal environment that characterizes pregnancy. Some of the obstetric complications that occur in women with adenomyosis in pregnancy include miscarriage, preterm delivery, placenta previa, low birth weight for gestational age, obstetric hemorrhage, and the need for cesarean section. These complications are an understudied field and remain unknown to the majority of obstetricians. These pathological conditions pose challenges to both the typical progression of pregnancy and the smooth conduct of labor in affected women. Further multicenter studies are imperative to validate the most suitable method for concluding labor following surgical intervention for adenomyosis.

## 1. Introduction

Adenomyosis refers to the presence of endometrial glands deeply embedded in the myometrium in a random arrangement. Similar histological lesions can also appear outside the uterus, such as in the area of the rectal septum and posterior and oior compartments [1,2].

The pathogenesis and etiology of adenomyosis are not yet fully understood. Adenomyosis may develop due to a “weakness” in the uterine cavity’s increased pressure, in the smooth muscle fibers of the myometrium, or both. Sustaining adenomyosis may require elevated levels of estrogen and compromised regulation of ectopic endometrial growth, a process potentially linked to immune system dysfunction. It is possible to cause fibrosis, leading to uterine enlargement and menorrhagia. It is difficult to diagnose clinically and is usually diagnosed retrospectively after a histological examination of the uterus. It is often considered a different entity from endometriosis and occurs in a different population with a distinct etiology. This is infiltration, not displacement, of the myometrium [3,4].

Hyperplasia and hypertrophy of smooth muscle fibers reflect the reactive changes associated with ectopic endometrial proliferation. While the definitive diagnosis is typically established post-hysterectomy, there have been efforts to confirm it preoperatively using magnetic resonance imaging (MRI) and endometrial biopsies [4,5].

## 2. Pathogenesis of Adenomyosis

Adenomyosis, a prevalent gynecological condition occurring in a range of 5% to 70%, can have a profound impact on women’s quality of life. Clinical symptoms frequently include menorrhagia, painful menstruation, and difficulties with fertility [6,7,8]. Two main theories exist regarding the origin of adenomyosis. One theory, known as “migration”, suggests that the endometrium penetrates into the myometrium. The other theory involves the metaplastic differentiation of endometrial stem cells that remain within the myometrium.

The detection of mutations primarily in the KRAS (Kirsten Rat Sarcoma virus) genes in cases of adenomyosis highlights the significant role of these genes in the genetic pathogenesis of the disease. This finding contradicts a recent theory suggesting that molecular abnormalities in adenomyosis are primarily epigenetic or linked to the abnormal expression of various genes [7,9]. The identification of KRAS gene mutations in adenomyosis challenges the recent theory that molecular abnormalities in adenomyosis are primarily epigenetic or linked to abnormal gene expression [9] (Figure 1). The most notable abnormalities in gene expression associated with adenomyosis involve excessive estrogen production and progesterone resistance and are linked to steroid hormone receptors and other transcription factors.

The *ESR1* (*Estrogen Receptor 1*) gene, which codes for the estrogen receptor alpha (ERa) and is found on chromosome 6q25.1, has been found to have mutations, the most common of which are P129R and M427I/L429M [9,10]. These mutations appear to be involved in the process of adenomyosis development. An association has been noted between adenomyosis and the disruption of mechanisms involved in the transition from epithelial to mesenchymal cells, evidenced by reduced expression of the cadherin-1 (CDH1) protein and elevated levels of Notch and TGF-β.

Epigenetic factors also appear to be implicated, with Class I histone deacetylases (HDACs) and DNA methyltransferase (DNMT) proteins potentially playing roles. Increased levels of HDAC1, HDAC3, DNMT1, and DNMT3B have been observed in cases of adenomyosis within the ectopic endometrium. These findings suggest that epigenetic alterations are significant contributors to adenomyosis pathogenesis, supplementing our understanding alongside the aforementioned genetic factors, and can help diagnose and treat the disease [9,10].

While the precise etiological factors of adenomyosis remain uncertain, various theories have been proposed. The prevailing notion suggests that adenomyosis arises from the attachment of the basal layer of the endometrium to the myometrium. In extraneous uterine regions, the primary theory regarding adenomyosis pathogenesis involves the de novo emergence of ectopic fetal remnants from Muller’s ducts, as no established invasive mechanism of the endometrium into the myometrium has been confirmed [11,12,13]. There are notable cellular distinctions between the basal and functional layers of the endometrium, including heightened DNA synthesis in the nucleus and margin formation in the functional layer. Typically, the functional layer serves as the site for blastocyst implantation, whereas the basal layer facilitates intrauterine regeneration following menstruation [14,15,16,17].

During endometrial regeneration, basal layer gland cells closely interact with endothelial cells possessing intracellular microfiber/tubular and squamous cell systems [18,19,20,21,22]. These findings suggest a possible mechanism of migration involving amoebic contraction-extension. Although such morphological changes have not yet been observed in the intrauterine epithelium of adenomyosis, in vitro studies indicate that endometrial cells possess penetration capabilities comparable to cell lines from metastatic bladder carcinoma. This penetration capacity may aid in the extension of the basal layer of the endometrium into the myometrium [23,24]. Additionally, the production of tenascin-c by endometrial layer fibroblasts, a fibronectin inhibitor that promotes epithelial cell migration, implies a complex physicochemical relationship during the proliferative phase of endometrial growth.

Tenascin production is also found in MCF-7 cells from breast cancer and is stimulated by epidermal growth factors (EGFs), which are regulated by hormones [25,26]. During the proliferative phase of the menstrual cycle, tenascin has been identified around endometrial glands through immunohistochemistry, but not in this location after ovulation. Tenascin likely mediates interactions between epithelial and mesenchymal cells, inhibiting cell adhesion to fibronectin in both endometrial adenomyotic and normal endometrium [27]. In a study utilizing in situ hybridization and immunohistochemistry, it was discovered that endometrial glands in adenomyosis exhibit selective upregulation of human chorionic gonadotropin (hCG) receptor mRNA and immunoreactive receptor protein compared to healthy tissue [26]. It appears that hCG/LH receptor expression levels do not vary among different sites within the normal endometrium.

However, increased expression of these receptors may grant epithelial cells the capacity to invade the myometrium and form adenomyotic lesions. Additionally, intriguingly, there is heightened expression of hCG/LH receptors in both endometrial carcinomas and non-invasive choriocarcinoma trophoblast cells [13,26]. Studies on steroid hormone receptors in adenomyosis lesions have yielded conflicting results. While some studies have reported the absence of progesterone receptors in 40% of adenomyosis cases, others have indicated higher concentrations of progesterone receptors compared to estrogen receptors. Immunohistochemical detection techniques have revealed relatively high concentrations of estrogen and progesterone receptors in both the basal and adenomyotic endometrium. Estrogen receptors are essential for uterine development, which is driven by estrogen [13,28].

While clear evidence of a disrupted hormonal environment in most women with adenomyosis is lacking, hyperestrogenemia may contribute to the process of endometrial infiltration, given the high incidence of endometrial hyperplasia among women with adenomyosis. Some researchers suggest that a relatively elevated concentration of estrogen is necessary for the development of both endometriosis and adenomyosis. Clinical observations supporting this hypothesis include the regression of ectopic endometrium and alleviation of associated symptoms, such as menorrhagia and dysmenorrhea, upon the destruction of the estrogenic environment with danazol [1,29]. Similar to uterine fibroids, adenomyotic tissues synthesize and secrete estrogen [1,29,30]. Aromatase activity of estrogen sulfatase was detected in the upper part of the myometrium containing adenomyosis foci, as revealed by steroid biochemical analysis. This activity, particularly that of aromatase, surpassed the levels observed in normal adjacent endometrium, leiomyomas, and suprauterine endometrium.

Adenomyotic tissue demonstrates a favorable response to progesterone, exhibiting secretory differentiation. Progestogens also increase aromatase activity in both ectopic endometrium and adenomyotic tissues, thereby contributing to estrogen biosynthesis in adenomyotic foci. However, it is plausible that the mere presence of sex steroids alone may not be adequate for the development of adenomyosis. It is conceivable that in cases of adenomyosis, the myometrium may be predisposed to invade the main endometrium, leading to benign “penetration” of the endometrium as a secondary occurrence due to a “weak” myometrium or as a consequence of uterine interventions, such as scraping, fibromyectomy, and cesarean section. For example, scraping one horn of the uterus and fallopian tube while retaining pregnancy in the opposite horn was used to induce adenomyosis in pregnant rabbits [31,32].

The penetration of the basal layer of the endometrium into the myometrium may be facilitated by elevated intrauterine pressure, possibly induced by high levels of circulating progesterone [33]. In cases of idiopathic endometriosis and adenomyosis, immunohistochemical research has shown enhanced expression of class II antigens of the major histocompatibility complex (HLA-DR) in glandular cells of the endometrium. Furthermore, there appears to be an elevated presence of macrophages in the myometrium of women with adenomyosis, which can activate helper T and B cells to produce antibodies. Phospholipid autoantibodies and the significant deposition of immunoglobulins (Igs) or complement factors have been detected in women with endometriosis or adenomyosis [30,33]. The precise significance of these immune irregularities in adenomyosis or endometriosis remains unclear.

Research conducted in vitro has shown that interferon γ, which is secreted by activated CD3+ T cells in the uterus, stimulates the expression of HLA-DR immunoreactivity in endometrial glandular cells and inhibits their growth. The proliferation of endometrial cells is more strongly inhibited the closer they are to activated T cells. Lymphocyte-like formations, predominantly located at the endomyometrial junction, are abundant in activated T cells. Their appearance correlates with the maximum suppression of endometrial growth, as observed morphologically and through proliferation indices. Conversely, endometrial proliferation is predominantly observed near the surface of the endometrium, away from the basal layer where these lymphoid formations are situated [34,35] (Figure 2).

## 3. Pathological Anatomy

During hysterectomy, the adenomyotic uterus typically exhibits a spherical or soft shape. Swelling is observed in approximately 60% of cases but rarely exceeds the size of the uterus at 12 weeks of pregnancy. The weight of the adenomyotic uterus ranges from 80 to 200 g. In a landmark study by Langlois, the upper limit of normal uterus weight was determined based on parity, with unmarried women having a limit of 130 g, first- to third-time mothers having a limit of 210 g, and women with four or more children having a limit of 250 g [36]. Following these criteria, excluding cases involving fibroids, the weight of the uterus does not significantly increase with adenomyosis.

Uterine fibroids with adenomyosis are usually hyperemic with thick walls, and researchers have reported that adenomyosis is more common in the posterior wall of the uterus than in the anterior [4]. Bird and colleagues discovered that the foci of adenomyosis were evenly distributed when six additional incisions were made for histopathological examination [37]. In the myometrium, these foci can occasionally be large and confined, creating formations known as adenomas, or they can be widely distributed. The myometrium, which surrounds the endometrium, is enlarged, giving adenomyosis its distinctive macroscopic appearance. When the entire myometrium, or one of the layers of the uterine wall, is diffusely affected by adenomyosis, the uterus enlarges and assumes a spherical shape. Upon cross-sectioning of the uterus, hypertrophic muscular beams are visible, extending in all directions and encircling the foci of adenomyosis. In some instances, these foci may contain “old” blood with a brownish appearance, indicative of hemolyzed blood and hemosiderin deposits [38].

Local infection of the uterus by adenomyosis can resemble fibroids. The term “adenomyoma” is commonly used to describe the frequent occurrence of adenomyosis. However, because the treatment does not involve neoplastic processes, Hendrickson and Kempson prefer the term “focal adenomyosis”. Adenomyoma is often clinically confused with leiomyoma, a benign but neoplastic condition; thus, the term “adenoma” is accepted. Typically, adenomas blend with the normal myometrial environment and lack clearly defined boundaries, whereas leiomyomas compress the myometrial environment and have well-defined boundaries. [39,40]. Leiomyomas can be nucleated, while adenomyomas cannot exhibit this feature. Histologically, through immunohistochemical analysis, the endometrial glands and the layer within the foci of adenomyosis resemble the basal layer of the endometrium. They rarely respond to hormonal stimuli, which partly explains why hemorrhagic or regenerative morphological findings are observed only in certain cases of adenomyosis foci.

The underlying cause of the heightened tendency for focal bleeding in deep-seated adenomyotic foci remains unclear [2,41]. On the other hand, similar to the functional layer of the endometrium, the ectopic endometrium at the endometriosis foci frequently experiences cyclic changes, such as degeneration, bleeding, and regeneration. The variable occurrence of changes in menstruation type could be explained by the ectopic endometrium’s low vascularity, which is indicative of a primary endometrial type, in contrast to the endometrium proper’s high vascularity, which forms the endometrium’s functional layer. Nonetheless, the ectopic endometrium retains the capacity to proliferate, which can lead to its development and be responsible for the persistence of amenorrhea or submenorrhea following endometrial destructive procedures [42].

Secretory changes, including layer degradation in adenomyosis foci, are predominantly observed during pregnancy and treatment with exogenous progestogens mediated through estrogen and progesterone receptors. The progesterone effect in the non-pregnant uterus is observed in approximately 30-50% of adenomyosis foci. Some authors have noted degeneration during pregnancy, particularly in deep foci located at least two low-magnification optical fields deep, while degeneration is absent or minimal in foci less than two low-magnification optical fields deep, near the basal layer of the endometrium–myometrium boundary [43,44]. It is important to note that adenomyosis can frequently be complicated by hyperplastic disorders progressing to atypia, while squamous cell carcinoma, mucosal metaplasia, and adenocarcinoma can occur concomitantly with adenomyosis.

When carcinoma is limited to an adenomyotic lesion, it is termed “intravenous”, and the prognosis is no worse than the carcinoma for which the patient underwent surgery. Histologically, it is challenging to ascertain whether adenocarcinomas found in the supracervical uterus within adenomyosis foci represent primary lesions or expansion of the endometrium into the adenomyotic foci [45].

## 4. Clinical Symptoms

Approximately 35% of adenomyosis cases are symptomatic [46,47,48,49]. In other cases, the most common symptoms are menorrhagia (50%), dysmenorrhea (30%), and uterine bleeding (20%). In some cases, discomfort may be an additional symptom. The frequency and severity of symptoms depend on the extent and depth of adenomyosis [46,47,48,49]. The exact cause of menorrhagia in patients with adenomyosis is unknown. Dysmenorrhea, a clinical condition characterized by painful menstrual cramps, is notably prevalent among adolescents and may serve as a crucial early indicator of endometriosis. Recognizing the correlation between dysmenorrhea and potential concomitant endometriosis is essential for early detection and intervention, which can mitigate long-term adverse health outcomes and enhance the overall well-being of affected adolescents [50].

Menorrhagia in adenomyosis may result from the poor contractility of the adenomyotic myometrium and the compression of the endometrium by submucosal adenomas or leiomyomas. Mefenamic acid can reduce blood loss, suggesting that prostaglandin F2a (PGF2a) may contribute to increased blood loss in women with adenomyosis [5,46,47]. Other contributing factors may include anovulation, hyperplasia, and, rarely, endometrial adenocarcinoma. Dysmenorrhea is attributed to irritability of the uterus, which is secondary to the increased amount of blood loss [48,49,50,51]. The symptoms associated with adenomyosis have not been uniformly explained by all researchers. For instance, in a study involving 136 patients with histologically confirmed adenomyosis, the symptoms were diverse and non-specific. According to the researchers, these symptoms were more likely linked to coexisting pathological conditions, such as leiomyomas, endometriosis, and polyps, rather than to adenomyosis itself [52,53]. Additionally, in another prospective study, there were no differences in the incidence or severity of dysmenorrhea and pelvic pain between 28 women with adenomyosis and 157 control subjects [54].

In a prospective study, patients with adenomyosis experiencing mild or no dysmenorrhea (n = 40, Group 1) and those with moderate to severe dysmenorrhea (n = 80, Group 2) were recruited. Immunohistochemistry (IHC) analysis was conducted in 60 cases to assess the cellular levels of estrogen receptor-α (ER-α), estrogen receptor-β (ER-β), gonadotropin-releasing hormone receptor (GnRH-R), and neurofilaments (NFs). The findings revealed that a history of cesarean section (CS) was positively associated with the severity of dysmenorrhea in adenomyosis (OR (95% CI): 4.397 (1.371–14.104)). Furthermore, ER-α levels in the eutopic endometrium (EUE) of Group 2 were higher compared to those in the ectopic endometrium (ECE) of Group 1. Additionally, Group 2 exhibited elevated NF levels in the ECE relative to the EUE (62).

## 5. Diagnosis

Given the non-specific nature of adenomyosis symptoms, it is unsurprising that the disease is rarely diagnosed preoperatively. Most researchers report a correct preoperative diagnosis in fewer than 10% of cases. However, due to factors such as case selection methods, incomplete pathological examination of surgical specimens and the limited number of well-designed studies accurately assessing the true diagnostic ability of adenomyosis remain challenging [55,56,57]. The clinical diagnosis of adenomyosis is, at best, hypothetical in about 50% of cases and often does not occur at all (75%) or is overdiagnosed (35%) [19,56,57].

Although symptoms such as dysmenorrhea and menorrhagia in women between the ages of 40 and 50 may suggest adenomyosis, they do not support the diagnosis. In certain situations, the uterus may be soft and sensitive to touch, and it may be diffusely expanded to the size of a 12-week pregnancy. The sole factor directly linked to adenomyosis is the existence of endometrial hyperplasia at the time of hysterectomy. Several researchers have employed radiological methods for diagnosing adenomyosis. In the largest hysterosalpingography study, Marshak and Eliasoph diagnosed adenomyosis in only 38 out of 150 patients with confirmed adenomyosis [58]. However, they did not report the total number of patients examined or the frequency of false-positive diagnoses.

The most common findings in hysterosalpingography are endometrial diversions and cellular invasions within the myometrium [59,60,61]. This test was considered inaccurate because myometrial adhesions attributed to adenomyosis resemble lymphatic or vascular infiltrations of the pigment. Intra-abdominal ultrasound is not useful for diagnosing adenomyosis. In the late 1970s, a group suggested that abnormal ultrasound areas of the myometrium, 5–7 mm in size, were characteristic of generalized adenomyosis [59,60,61]. However, this view was later challenged by Siedler et al., who reported that most women with established adenomyosis exhibited generalized uterine enlargement, normal uterine echogenicity, and retained uterine shape.

Subsequent studies have not resolved this issue [62]. Transvaginal ultrasound has been utilized to diagnose adenomyosis since the early 1990s. Using a preoperative transvaginal ultrasound, Fedele evaluated 43 women who were scheduled for hysterectomy due to menorrhagia. He identified numerous small hypoechoic areas within the myometrium, with an abnormal ultrasound outline in 22 of these women [63]. The method’s sensitivity was estimated at 80%, and its specificity at 74%.

However, other researchers reported lower sensitivities, at 48% and 53% [63,64,65]. To better understand this issue, larger research studies with a greater number of women are required. Additionally, pelvic pathology has been the subject of magnetic resonance imaging (MRI), with promising early results in women with adenomyosis [5,66]. Mark and colleagues identified adenomyosis in 8 out of 20 women using T2-weighted images. Of the remaining 12 women, ten were correctly diagnosed as free of adenomyosis, while the diagnosis was uncertain in two cases [67]. The researchers described a typical, wide, low-intensity area surrounding the normal, high-intensity endometrium in women with diffuse adenomyosis, although tiny foci of adenomyosis could not be detected. T2-weighted imaging demonstrated significant advantages over T1-weighted imaging, especially without contrast enhancement. MRI has been employed to differentially diagnose adenomyosis from leiomyomas [5,66]. In a related study, 93 patients were preoperatively evaluated, and the results were correlated with surgical pathology findings. Seventeen cases of adenomyosis were accurately diagnosed preoperatively [68].

Nevertheless, the broader application of this new technology requires further assessment [69,70]. Additionally, the high cost may limit the widespread use of MRI as a diagnostic tool [71].

CA-125 is an antigen produced by ovarian epithelial cells, secreted into the bloodstream and associated with various gynecological conditions. Researchers have utilized CA-125 level measurements to predict recurrences of non-mucosal ovarian cancers and to diagnose endometriosis recurrences, with the latter requiring successive measurements [72,73].

In 1985, Takahashi and colleagues reported elevated preoperative CA-125 levels in six out of seven women with adenomyosis [74]. Although these levels were raised, they were significantly lower than those observed in ovarian cancer patients. One month post-hysterectomy, CA-125 levels returned to normal in all women. Immunohistochemical methods showed CA-125 production in the glandular epithelium of adenomyosis foci in eight hysterectomy specimens [75,76]. In contrast, Halila and colleagues were unable to replicate these findings. In a study of 22 women, 11 of whom had adenomyosis, they found normal preoperative CA-125 levels in all women with adenomyosis, with no significant changes one and five weeks post-surgery. The reasons for these differing results remain unclear, necessitating further research for clarification [77,78].

Cysteine and leucine aminopeptidase levels have also been considered potential markers for adenomyosis, as their levels are elevated in various benign and malignant uterine and ovarian conditions [79]. However, control studies are lacking to evaluate the clinical utility of these measurements. While adenomyosis can be diagnosed through uterine needle biopsy, this method has low sensitivity, depending on the number of biopsies and the depth of adenomyotic penetration. This technique is not particularly useful for diagnosing minimal or moderate disease, but can provide histological confirmation in cases of extensive myometrial infiltration. If a biopsy confirms adenomyosis, patients should be identified based on their history and undergo transvaginal ultrasound and MRI, which can also help guide biopsy location. However, routine myometrial biopsies should not be performed in women with pelvic pain [80,81,82,83].

## 6. Adenomyosis and Pregnancy

Adenomyotic tissue is affected by hormones, making it susceptible to hormonal fluctuations associated with pregnancy. Ectopic endometrium in patients with adenomyosis does not provide an ideal environment for placentation and leads to placental disturbances resulting in preterm delivery, restricted intrauterine growth, and small-for-gestational-age neonates at delivery. There are several reasons why adenomyosis may contribute to negative pregnancy outcomes [84,85,86,87] (Figure 3).

Specifically, the endometrium presents resistance to the selective effects of progesterone, which typically induces stromal cell transformation and an embryo-receptive phenotype. Due to this resistance, critical genes for embryo implantation, such as FOXOA (prolactin), IGF-II (fertilization response), and glycodelin, are dysregulated in the endometrium of affected embryos [88,89,90].

Additionally, adenomyosis may be associated with adhesion formation, either due to the disease itself or as a result of surgical intervention. Adhesions can put tension on nearby structures during pregnancy because of the uterus’s growth. Another reason adenomyosis leads to adverse pregnancy outcomes is the inflammatory processes it triggers at both the endometrial and systemic levels. Maternal systemic inflammation and placental ischemia are causes of preeclampsia or premature birth. Also, endometriotic tissue tends to penetrate into the walls of vessels and tissues, predisposing their rupture.

During the menstrual cycle, uterine contractions happen, regardless of the existence of pregnancy. The contractions seen during the menstrual cycle are called endometrial waves and involve only the sub-endometrial layer of the myometrium. Compared to women without adenomyosis, patients with the condition exhibit uterine contractions that are more frequent, are wider, and have a greater tone [88,89,90]. Studies have shown that women with adenomyosis experience uterine contractions that are more frequent, are wider, and have a greater tone compared to those without adenomyosis.

It has also been demonstrated that women with adenomyosis experience an increased release of reactive oxygen radicals and heightened expression of enzymes that result in the accumulation of these radicals in their cells.

Maternal endothelium dysfunction is known to be largely caused by oxidative stress, and this dysfunction ultimately results in unfavorable obstetric outcomes. In comparison to women without adenomyosis, women with the disorder have been found to have a thicker junctional zone, which is the highly specialized inner third layer of the myometrium.

Normal placentation involves the full transformation of spiral arteries into large vessels that link the placenta and uterus at the junctional zone. In contrast, abnormal placentation is marked by the absence or incomplete remodeling of these arteries [90,91]. (Figure 3: Pathological impact of adenomyosis on pregnancy.)

Research on the pathogenesis of poor pregnancy outcomes indicates that all of these local endometrial alterations may be associated with a large number of obstetric complications [84,85,86,87,92,93] (Table 1).

### 6.1. Abortion Miscarriage

A miscarriage is defined as the termination of a pregnancy before 20 weeks of gestation, and it is classified as “recurrent” if three or more occurrences take place. During the first trimester of pregnancy, miscarriage is the most prevalent consequence [94].

The relationship between adenomyosis and miscarriage has been the subject of several studies. One study found that the miscarriage rate was significantly higher in women with adenomyosis, with an odds ratio (OR) of 2.11 (95% CI 1.33-3.33). Surgical treatment for adenomyosis has been shown to increase natural conception rates, although the data are limited to only two studies. Conversely, treatment with GnRH analogs does not appear to improve IVF outcomes in women with adenomyosis. In fact, women with adenomyosis generally have poorer IVF clinical outcomes, and pretreatment with long-term GnRH analogs has not proven beneficial, as indicated by a meta-analysis that included only three studies.

Additionally, many studies fail to distinguish between focal and diffuse adenomyosis, which introduces a significant source of bias [95,96]. Another study highlighted that the prevalence of adenomyosis was significantly higher in women with recurrent miscarriages (RM) compared to healthy controls. In this context, adenomyosis, along with uterine anomalies, has emerged as one of the most common risk factors associated with RM [97]. Furthermore, adenomyosis has been linked to decreased outcomes in assisted reproductive technologies (ART), increased miscarriage rates, and possibly recurrent pregnancy loss (RPL) [98,99,100]. In contrast, another study suggested that while implantation is not impacted by adenomyosis, increased miscarriage rates result in lower-term pregnancy rates, highlighting a clear negative effect on the final outcome of oocyte donation (OD) [101].

### 6.2. Preterm Birth

Delivery prior to the 37th week of pregnancy is known as premature birth, a leading cause of morbidity in newborns and long-term health issues in adults. It continues to be a significant concern in obstetrics, impacting 5% to 15% of all pregnancies. Preterm delivery is strongly associated with cardiovascular, renal, and pulmonary disorders, diabetes mellitus, cancer, and obesity in adulthood. It also increases the risk of serious neonatal conditions such as intra-abdominal hemorrhage, necrotizing enterocolitis, and respiratory distress syndrome. Known causes of premature birth include multiple pregnancies, polyhydramnios, sexually transmitted infections, and smoking. Many premature births, however, remain idiopathic or occur following obstetric complications, such as preeclampsia. Additionally, infants conceived through assisted reproductive technologies (ART), even in singleton pregnancies, have a higher likelihood of being born preterm. The exact etiology remains unclear [102].

Fernando et al. demonstrated that the rate of preterm delivery in infertile patients with ovarian endometriomas who underwent IVF is twice that of the general population [103]. This phenomenon appears to be attributable to the distinct molecular pathology of ovarian endometriomas compared to peritoneal endometriosis [104,105]. The relative risk for preterm delivery even less than 28 weeks and for premature rupture of membranes in women with adenomyosis is twice as high compared to the general population [106]. Harada and colleagues demonstrated that women with adenomyosis and those with endometriosis are at a high risk for premature rupture of membranes and preterm delivery, irrespective of IVF techniques.

In fact, pregnant women with adenomyosis have a twofold increased risk of preterm delivery and premature fetal membrane rupture compared to those with endometriosis [107]. Also, in preterm labor, as in adenomyosis, progesterone receptor activator hypermethylation and an increase in progesterone isoform A (reduced uterine responsiveness to progesterone) occur. The above events lead to pathological contractility of the uterus and alterations of the junction zone (JN)/myometrium of the uterus in women with adenomyosis [98]. In women with adenomyosis, preterm labor and premature rupture of fetal membranes can be explained by the failure of the normal transformation of the spiral arteries in the inner myometrial segment, called the JZ junction zone. Placental hypohydration seems to increase the production of pro-inflammatory mediators, resulting in local and systemic inflammation and, finally, the onset of labor [108].

### 6.3. Ectopic Pregnancy

The implantation of a fertilized egg outside the uterine cavity is the hallmark of an ectopic pregnancy. The fallopian tubes, for example, are the site of ectopic pregnancy in about 95% of cases [109].

Regarding adenomyosis, while its coexistence with infertility is suggested, a definitive causal relationship has not been fully established. Adenomyosis has been associated with various adverse reproductive outcomes, including ectopic pregnancy [110]. Adenomyosis is one of the factors that predispose individuals to the development of intramural pregnancy [111]. Intramural ectopic pregnancies, although exceedingly rare, are believed to occur when a developing pregnancy travels through a tract from an ectopic endometrial rest into the myometrium, resulting in intramural implantation [112].

### 6.4. Small for Gestational Age (SGA)

Newborns whose birth weight is small for gestational age are those who weigh less than the 10th percentile on the fetal growth curve [113]. Harada and colleagues elucidated that neonates born with low birth weight for gestational age are more associated with adenomyosis than with endometriosis [108]. Women with endometriosis or adenomyosis had higher odds of preterm delivery and delivering a child small for gestational age (SGA) compared to women without these conditions, with the highest odds observed among women with adenomyosis [114,115]. Blood flow within the adenomyosis lesions is abundant, while the placenta has reduced blood flow based on the results of blood flow measurements in the myometrium and placenta of women with adenomyosis and severe SGA. Furthermore, chronic inflammatory processes in the uterine microenvironment and uterine resistance to progesterone may be the possible causes of SGA [114,115,116].

### 6.5. Fetal Growth Restriction (FGR)

Fetal growth restriction is defined as the pathological reduction in fetal growth rate that results in a fetus/neonate that does not grow as expected for gestational age and is at risk of increased perinatal morbidity and mortality. In the majority of studies, a fetus is defined as a fetus weighing below the 3rd percentile or below the 10th percentile on the fetal growth curve with abnormal flow Doppler in the middle cerebral and umbilical or uterine arteries. Normally, resistance to blood flow is greater in the middle cerebral artery than in the umbilical artery [117]. Many studies have shown that women with a histopathologic diagnosis of adenomyosis have an increased prevalence of fetal growth restriction (FGR) [114,115,116,117,118].

### 6.6. Hypertensive Disease of Pregnancy

Gestational hypertension is defined as a persistent (elevated blood pressure in at least 2 readings 6 h apart) occurrence of blood pressure > 140/90 mm Hg that developed after 20 weeks of gestation in a previously normotensive woman without the presence of albuminuria [119]. In 2022, a study underscored the connection between adenomyosis and an elevated risk of hypertension, emphasizing the importance of vigilant blood pressure monitoring for affected women [120]. Additionally, another study revealed that a larger adenomyosis size and the diffuse type are associated with unfavorable pregnancy outcomes. Women with diffuse-type adenomyosis should receive close monitoring due to increased occurrences of pregnancy-induced hypertension and uterine infections [121].

### 6.7. Preeclampsia

Preeclampsia is defined as the occurrence of arterial hypertension and proteinuria, >300 mg/24 h, during pregnancy [122]. It is a clinical syndrome with an increased fetal and maternal mortality rate and affects approximately 5% of all pregnancies. In women with severe preeclampsia, only 10% of the spiral arteries are altered, resulting in pathological placentation, with increased resistance vessels, with activation of the coagulation mechanism and endothelial cell dysfunction. Abnormal placentation could cause abnormal infiltration of the trophoblast. It is generally considered that a uterine junctional zone >12 mm thick is a diagnostic criterion for adenomyosis.

The uterine junction zone of women with adenomyosis >8 mm was thicker than that of women without adenomyosis 5–8 mm. There is a positive correlation between the thickness of the junctional zone of the uterus and the stage of adenomyosis. Because of the different thicknesses of the uterine junction zone between women with endometriosis and women with adenomyosis, the types of obstetric complications observed in pregnant women with endometriosis and women with adenomyosis were different [123]. In Harada’s meta-analysis, the adjusted OR of mild preeclampsia in women with adenomyosis regardless of ART treatment was found to be 1.76. Thus, women with adenomyosis have an increased risk factor for preeclampsia [108].

Furthermore, a 2016 study revealed that adenomyoma is commonly found in patients with preeclampsia complicated by fetal growth restriction. However, this study indicated that there is no overall association between preeclampsia and adenomyosis. A comparison involving patients with late-onset and early-onset preeclampsia, supplemented by MRI findings, highlighted a significant correlation between late-onset preeclampsia and indirect markers of adenomyosis [124].

### 6.8. Gestational Diabetes

Gestational diabetes is defined as carbohydrate intolerance, with onset during pregnancy and with a positive glucose tolerance test [125]. Adenomyosis has been associated with increased occurrences of gestational diabetes [126]. Moreover, a systematic review identified a single paper on gestational diabetes mellitus included in this study, thus presenting it once. Despite indications of elevated diabetes mellitus risk, a case-control study did not reveal a significant increase in the likelihood of gestational diabetes [100,127].

### 6.9. Placenta Previa

The placenta separates from its implantation site prior to delivery, a condition known as placental abruption, which is typically linked to placenta previa. Placenta previa, which affects 0.5 to 0.3% of pregnancies, is the result of an aberrant placental implantation in the lower region of the uterus, covering part or all of the internal cervical os.

In the second or third trimester of pregnancy, it presents clinically as painless vaginal bleeding. When there is significant bleeding, there is a higher risk of morbidity and mortality for both the mother and the newborn [128]. A study conducted by Harada and colleagues showed that placenta previa is a very common complication in women with a history of endometriosis, but less common in women with adenomyosis, regardless of their age and the method of conception.

Many studies have shown that adenomyosis in pregnancies is associated with an increased risk of placenta accreta and placental abruption [116,118,126]. This is anticipated, as when the placenta is implanted in the lower uterine segment as opposed to the proximal sites, the trophoblast can more readily penetrate the myometrium [86].

### 6.10. Obstetric Bleeding

Postpartum hemorrhage (PPH) is defined as the loss of more than 500 mL or 1000 mL of blood within the first 24 h postpartum after natural delivery or cesarean section, respectively [86,129]. In the 2009 study by Healy and colleagues, postpartum hemorrhage had an increased risk of occurrence in patients with adenomyosis (OR, 1.3 95% CI, 1.1–1.6) compared with those without adenomyosis. In the same study, fresh oocyte transfer in artificial cycles was associated with increased obstetric bleeding (OR, 1.8 95% Cl, 1.3–2.6) compared with fresh oocyte transfer in natural cycles [130]. It seems that there are some pathophysiologic mechanisms by which adenomyosis predisposes patients to postpartum hemorrhage [131].

### 6.11. Cesarean Delivery

Cesarean delivery is a natural consequence in patients with adenomyosis because other obstetric complications are statistically significantly increased [98,100,112,113,114,115,116,117,118,119]. Cesarean delivery is frequently indicated in patients with adenomyosis due to associated obstetric complications. These complications, including abnormal uterine bleeding, preeclampsia, and placenta previa, heighten the risk of adverse outcomes in vaginal delivery, thereby making cesarean delivery a safer and more controlled alternative for ensuring maternal and fetal well-being.

## 7. Surgical Treatment of Adenomyosis

The laparoscopic methods for treating adenomyosis include:Laparoscopic thermal destruction of the myometriumLaparoscopic adenomyomectomyLaparoscopic total or subtotal hysterectomy

Additionally, other treatment approaches have been proposed, such as hysteroscopic endometrial excision to alleviate symptoms, uterine artery embolization, MRI-guided focused ultrasound (MRgFUS), which is currently under investigation, and various medications [126,127,128,129,130,131,132].

### 7.1. Laparoscopic Thermal Destruction of the Myometrium

The thermal destruction of the myometrium aims to shrink adenomyosis by inducing necrosis. This technique has been tested for both diffuse and focal adenomyosis. However, it is less precise than surgical excision because the destruction area is not well controlled, and the effect within the myometrium may be insufficient. This limitation cannot be verified intraoperatively by the surgeon. Additionally, it may weaken the myometrium due to the replacement of pathological adenomyotic tissue with scar tissue. The extent of scar tissue formed through thermal destruction can be greater than that resulting from surgical excision.

Diffuse multifocal thermal damage to the myometrium containing foci of adenomyosis may be effective in treating clinical symptoms. However, excessive use of electrocautery can lead to uterine rupture during pregnancy. Therefore, this method is primarily recommended for women over 40 who do not wish to conceive.

Thermal destruction of the myometrium can be performed using either unipolar or bipolar needles with a power setting of 50 W. To avoid causing necrosis in the uterine serosa and the potential formation of postoperative adhesions, the generator is activated only after the needles are correctly placed within the myometrium. Electrocautery is applied at intervals of 1–2 cm. This method can be combined with hysteroscopic endometrial ablation to enhance treatment efficacy [132].

### 7.2. Laparoscopic Adenomyomectomy

This procedure is recommended when adenomyosis is focal and does not involve a large portion of the myometrium. It is also suitable for women of reproductive age who wish to preserve their fertility. Although preoperative administration of GnRH analogs or danazol can reduce uterine vascularity, correct potential anemia, and minimize intraoperative bleeding, it is not advised because it can obscure the identification of the affected area during surgery [133].

The surgical steps for adenomyomectomy are similar to those for myomectomy and include:Cross-section of the lesion using unipolar diathermy.Preparation of the adenomyoma with meticulous hemostasis.Suturing the uterine wall in one or two layers or placing a “double” seromuscular suture in the uterus.Removal of the adenomyoma with an endoscopic morcellator.

Adenomyosis involves infiltrative myometrial damage without a clear cleavage plane. Unlike fibroids, the myometrium is infiltrated rather than displaced. This technical challenge is present in both open and laparoscopic approaches [134].

However, laparoscopic adenomyomectomy has distinct surgical characteristics that set it apart from myomectomy due to the nature of the condition:Infiltrative Nature: Adenomyosis infiltrates the myometrium without a cleavage plane, unlike in myomectomy. This requires the surgeon to “create” a surgical plan by excising tissues within the adjacent healthy myometrium, making it challenging to recognize the lesion.Tissue Texture: The glandular element with cystic areas and absence of a fibrous element in adenomyosis complicates firm capture and traction of the adenomyoma. Therefore, bidentate grasping forceps are often necessary instead of monodentate ones.Rich Blood Supply: Adenomyomas have a rich blood supply, necessitating meticulous hemostasis during removal. Bipolar diathermy is typically required, but the power setting should be low (~40 W), with short, repeated activations to avoid extensive scar tissue formation at the surgical margins.

To avoid the creation of an extensive zone of scar tissue at the sutured edges, careful technique is essential. After removing the adenomyoma from the myometrium, a tissue deficit is evident, which differentiates adenomyomectomy from myomectomy. The subsequent laparoscopic suturing must be strong and must maintain adequate perfusion of the sutured walls. This is particularly challenging when approximating the edges of the healthy myometrium, especially after extensive tissue removal.

Partial laparoscopic adenomyomectomy is recommended when the extent of myometrial infiltration by adenomyosis is so extensive that complete lesion removal is not feasible, ensuring that the uterus remains functional postoperatively. This “tumor-reducing” procedure aims to alleviate clinical symptoms [132,133,134,135].

### 7.3. Laparoscopic Partial or Subtotal Hysterectomy

Adenomyosis may increase complication rates associated with laparoscopic total or subtotal hysterectomy due to increased weight, increased vascularization, and impaired myometrial tissue of the uterus [136].

If fertility is not needed and the uterus is more than one-third covered in adenomyosis, it might be simpler to remove the upper half or two-thirds of the uterus while leaving any healthy myometrium in place. If one wishes to avoid menstruation, the remaining endometrium can be removed easily. Subtotal hysterectomy is better than partial hysterectomy when treating significant adenomyosis, because it is more likely to cure menorrhagia and dysmenorrhea and less likely for adenomyosis to return. The following are drawbacks of subtotal hysterectomy compared to total hysterectomy: a minor chance of recurrent or residual cervix adenomyosis, challenges in removing associated rectovaginal adenomyosis or adenomyoma, especially if it is attached to the cervix, and potential development of abnormalities in the cervical region that might necessitate additional surgery [132,133,134,135].

Subtotal hysterectomy may provide the following benefits over total hysterectomy: shorter recovery periods, less blood loss, quicker hospital discharge and return to regular activities, decreased risk of ureteric and bladder injuries, and decreased potential for negative consequences on sexual and bladder function.

Controlled trials comparing total and subtotal hysterectomy have not shown any significant differences in sexual function and have primarily focused on differences in operative and early postoperative events. Additionally, the number of these trials is too small to determine whether bladder and ureteric trauma differ, and the surgeons conducting the trials have not shown comparable efficacy in the two surgical techniques before beginning the trial [132,133,134,135,136,137].

## 8. Pregnancy Outcome—Rupture of Uterus

The postoperative pregnancy rate after laparoscopic thermal destruction of the myometrium or laparoscopic adenomyomectomy varies between 17.5% and 72.7%. However, artificial reproductive technology largely contributes to the relatively high pregnancy rate (89.8%) and the 80.8% labor rate.

Uterine rupture is a significant obstetric complication, particularly in patients with a history of endometriosis surgery. Cases reported in the literature typically involve uterine damage either immediately postpartum or during pregnancy, presenting as acute abdomen or obstetric bleeding. The rupture is often located in the posterior uterine wall. Notably, all reported cases have resulted in the delivery of healthy babies, without maternal fatalities.

Adenomyosis is significantly associated with the placenta accreta spectrum, including placenta accreta, placenta increta, and placenta percreta [138].

Furthermore, women who have undergone surgical treatment for uterine adenomyosis face a higher risk of spontaneous uterine rupture during subsequent pregnancies compared to those without a history of surgery, and complications such as placenta accreta spectrum frequently accompany uterine rupture. The incidence of uterine rupture is notably higher following the surgical removal of adenomyosis, with a literature review suggesting a 6% risk. This risk is significantly greater than the 0.26% risk following a myomectomy. Additionally, complications such as placenta accreta, placenta increta, and placenta percreta frequently accompany uterine rupture.

The frequency of uterine rupture in non-scarred uteri is approximately 0.005%, or 1 in 17,000–20,000 deliveries, but this increases to 0.04–0.2% in women with scarred uteri. After a previous cesarean section, the risk of rupture during vaginal birth rises to 0.27–0.7%. Pregnancies within 24 months of surgery are associated with a 2–3 fold increased risk of uterine rupture. Postoperative administration of GnRH analogs, LNG-IUD, or contraceptives is not recommended due to the increased risk of complications [135,136]. The postoperative fertility outcomes following surgical treatment of adenomyosis are summarized in Table 2 [139,140,141,142,143,144,145,146,147,148].

## 9. Obstetric Complications and IVF

Pregnant women with a history of endometriosis and adenomyosis, regardless of whether they conceived after ART treatment, have an increased risk of obstetric complications [135]. Regarding the risk of miscarriage, which is a frequent complication in women with endometriosis, most studies show that the risk of miscarriage does not increase in women with endometriosis who have undergone IVF [96,97]. In addition, IVF does not increase the risk of preterm birth in women with adenomyosis [102]. Also, newborns who at delivery have a small weight for gestational age are born twice as often from patients with ovarian endometriomas who performed IVF compared to the general population [108]. Another complication that occurs more frequently in women with adenomyosis who conceived through IVF than in those who conceived naturally is placenta previa [128].

On the other hand, it is worth noting that the method of conception, i.e., carrying out normal conception or carrying out in vitro fertilization, does not affect the incidence of complications such as obstetric bleeding, cesarean section, and preeclampsia in women with adenomyosis [112,123].

## 10. Discussion

Adenomyosis is a gynecological illness that is treatable yet incurable. It is a long-term, inflammatory, histogen-dependent condition marked by the presence of a layer of cells outside the uterus cavity and glandular endometrium. It primarily impacts women of reproductive age and a significant portion of the female population. While it can cause various clinical symptoms, the most prevalent ones are infertility, dysmenorrhea, and chronic pelvic pain. [111,117,130,131,132,133,134,135,136,137,138].

It is characterized as an enigma disease since, even today, the exact pathophysiology of the disease has not been ascertained. Its treatment is symptomatic, and it is necessary that the wishes of the patient, both in relation to her daily life and her quality of life and in relation to the will to have children, are taken very seriously. In recent years, evidence has emerged for the effect of endometriosis not only on the reduction of fertility but also on its effect on the outcome of pregnancy [115,116,117,118,138,139,140,141,142,143,144,145,146,147,149]. In the past, the dominant view was that pregnancy has a positive effect on endometriosis, adenomyosis, and its painful symptoms, not only due to the inhibition of ovulation that prevents bleeding of the endometrial tissue but also the metabolic, hormonal, immunological, and angiogenic changes associated with pregnancy [118,119,120,121,122,123,124,125,138,139,140,141,142,143,144,145,146,147,149].

However, current data on the role of adenomyosis in the course of pregnancy and its outcomes, as well as the effect of pregnancy on adenomyosis, diverge from the primary view. Today, several studies support a high correlation of adenomyosis with obstetric complications and emphasize the need to investigate these complications in order to prevent and improve the counseling of women with adenomyosis.

Results of published studies agree on few obstetric complications. [118,119,120,121,122,123,124,125,126,127,128,129,130,131]. The majority of studies suggest a high association of adenomyosis with spontaneous abortion, preterm delivery, small-for-gestational-age infants at delivery, placenta previa, ectopic pregnancy, and cesarean section. Furthermore, there is no evidence that prophylactic hormone therapy or surgery will prevent the adverse effect of adenomyosis itself on pregnancy outcomes. The results of studies on other obstetric complications are often controversial [133,134,135,136,137,138,139,140,141,142,143,149].

Complications such as preeclampsia, hypertension, and gestational diabetes in women with adenomyosis during pregnancy are rare. Acute complications of adenomyosis during pregnancy, such as spontaneous hemoperitoneum, bowel perforation, and possible rupture of the viscera, represent rare but life-threatening conditions that require, in most cases, surgical interventions. Due to the unpredictability of these complications, no specific recommendation for additional interventions is recommended in the routine pregnancy monitoring of women with a known history of adenomyosis. Regarding IVF in women with endometriosis, existing studies show that women with endometriosis who conceive with IVF have an increased chance of placenta previa compared to women with endometriosis who conceive naturally [121,147,148,150,151].

All of the above complications are an object that has not been sufficiently studied and remains unknown to the majority of obstetricians. There is no evidence that endometriosis disease has a significant negative effect on pregnancy outcomes. Therefore, pregnant women with adenomyosis can be reassured about the course of their pregnancy, although doctors should be aware of the possibility of an increased risk of certain complications [118,119,120,121,122,145,146,147,148,150,151].

Adenomyosis is a benign condition, which, located in the myometrium, causes an architectural disturbance in the layers of muscle fibers in a way that disturbs the cohesion, but also the strength of the uterus, especially in the case of pregnancy. Certainly, its diffuse form is a major problem in terms of treatment. Most often, this form of adenomyosis, upon failure of conservative methods, is an indication for hysterectomy. Laparoscopic thermal destruction of the myometrium by sites causes extensive zones of scar tissue within the myometrium, again resulting in disruption of the architecture and reduced strength of the uterus, making a future pregnancy unsafe. Laparoscopic adenomyomectomy, in the case of focal adenomyosis, is considered as safe as the corresponding one through laparotomy, as long as the surgeon has the necessary familiarity [120,121,122,123,124,148,150,151,152].

The tissue-filling deficit resulting from the exclusion of the adenomyoma is due to the pathogenesis of the disease itself, in which the endometrium is infiltrated and not displaced, as in the case of a fibroid. This technical difficulty occurs in both open and laparoscopic approaches postoperatively; both in the case of laparoscopic thermal destruction of the myometrium and in the case of laparoscopic adenomyomectomy, a test with magnetic tomography and color Doppler ultrasound may follow to evaluate the result, and also to assess the extent of the scar tissue and the possible location of remaining adenomyosis near it, before attempts at capture. With pregnancy, the problem remains the same, whether with open or laparoscopic surgery [143,144,145,146,147].

The removal of a large part of the affected myometrium or its extensive thermal destruction, according to positions, could be unsafe in a possible future pregnancy, due to, on the one hand, the reduction of the total volume of the uterus, and, on the other hand, due to the creation of scar tissue, which, causing tissue disruption cohesion, increases the chances of uterine rupture. However, the removal of a relatively well-demarcated adenomyoma, combined with diligent suturing of the healthy myometrium and reduced use of energy, may precede a future pregnancy as a safer surgical option than thermal destruction of the myometrium. Laparoscopic treatment of adenomyosis, whether it is diffuse or focal, is completely feasible and offers advantages to the patient compared to treatment through laparotomy [120,121,126,127,128,129,130,131,132,133,134,135,136,137,138].

Of course, it is a surgical approach that requires special familiarity on the part of the surgeon. Indications of the surgical options offered are laparoscopic thermal destruction of the myometrium by positions, laparoscopic adenomyomectomy, partial laparoscopic adenomyomectomy, laparoscopic total hysterectomy, and laparoscopic hysterectomy, which are functions of the extent of the condition and the clinical symptomatology [124,125,126,127,128,129,130,131,132,133,134,135,136,137,138]. In a small cohort of women diagnosed with adenomyosis prior to pregnancy, there was a significantly higher risk of cesarean section, preterm delivery, and postpartum hemorrhage [150]. Moreover, according to a recent retrospective case-control study [151], adenomyosis has been linked to a higher risk of placental cancer, preeclampsia, and miscarriages in the second trimester.

The type of adenomyosis may influence pregnancy outcomes, as evidenced by the higher rates of pregnancy-induced hypertension and breast infection in patients with diffuse-type adenomyosis compared to those with focal adenomyosis. Moreover, the same study found that the prevalence of cervical dysfunction increased with the extent of adenomyosis [148]. A recent prospective birth cohort study revealed that adenomyosis was a risk factor for preterm delivery (<37 weeks) and low birth weight of neonates (<1500 g) [152]. Regarding the pathogenic mechanisms involved in obstetric complications in adenomyosis, the role of inflammation, increased production of myometrial prostaglandins, alteration of uterine contractility, and intrauterine pressure has been implicated to a large extent [153].

In patients with adenomyosis, the activation of local and systemic inflammatory pathways was observed, which affects the matrix–trophoblast interactions as well as the dermis-permanent interactions, which may contribute to the activation of the mechanisms of preterm labor [153,154]. However, further research protocols are needed to precisely investigate the molecular mechanisms relating adenomyosis to obstetric complications. Adenomyosis has a negative outcome on IVF success rates, regardless of good oocyte and embryo quality. Specifically, it is reported that implantation rates, clinical pregnancy per cycle, clinical pregnancy per embryo transfer, and live birth rates in women with adenomyosis who underwent IVF/ICSI were significantly lower compared to those without adenomyosis. Additionally, the miscarriage rate is also higher in women with adenomyosis.

Several hypotheses have been proposed to explain the impact of adenomyosis on implantation, including alteration of the zona interstitium, increased uterine peristalsis, altered endometrial–myometrial vascular development, increased prostaglandin levels in the ectopic endometrial epithelium, increased expression of cytochrome cP450 aromatase in ectopic endometrium, lack of expression of adhesion molecules, and reduced expression of implantation factors. Furthermore, pregnant women with adenomyosis are reported to have an increased risk of obstetric complications. In summary, adenomyosis negatively affects the clinical outcomes of IVF, leading to reduced clinical pregnancy rates, full-term pregnancy rates, and higher miscarriage rates.

## 11. Conclusions

Finally, it is important to emphasize the need for more studies to assess whether there is a need to intensify pregnancy monitoring in women with adenomyosis. Studies should also explore the development of a specific protocol of ovarian stimulation for infertile patients with adenomyosis who wish to conceive. Additionally, before the initiation of the childbearing process, there is a need to investigate its utility, effectiveness, and ability to avoid obstetric complications. Adenomyosis is a benign condition, which, located in the myometrium, causes an architectural disturbance in the layers of muscle fibers in a way that disturbs the cohesion but also the strength of the uterus, particularly during pregnancy.

Certainly, its diffuse form poses significant challenges in treatment. Regarding pregnancy, the problem remains the same, whether with open or laparoscopic surgery. Due to the reduction of the total uterine volume and the creation of scar tissue, which disrupts tissue cohesion and increases the risk of uterine rupture, the removal of a large part of the affected myometrium or its extensive thermal destruction, according to positions, could be unsafe in a possible future pregnancy.

However, the removal of a relatively well-demarcated adenomyoma, combined with careful suturing of the healthy myometrium and reduced use of energy, may precede a future pregnancy as a safer surgical option than thermal destruction of the myometrium. The choice of surgical options depends on the extent of the disease and the clinical symptoms. Conservative interventions aimed at preserving fertility in patients with adenomyosis are evaluated based on symptom relief and obstetric outcomes. These patients should be informed about the higher chances of spontaneous abortions, ectopic pregnancy, and premature birth compared to the general population. More multicenter studies are needed to achieve safer results and reduce complications.

## Figures and Tables

**Figure 1 biomedicines-12-01925-f001:**
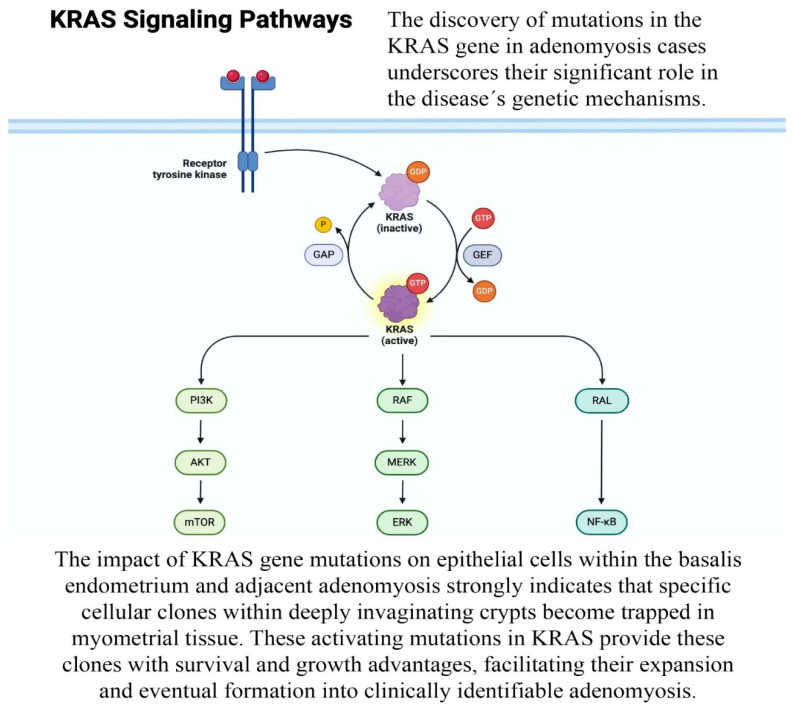
KRAS signaling pathways. KRAS pathway is activated by a receptor tyrosine kinase and regulates the intracellular pathways of mTOR, ERK and NF-kB. GAP: GTPase activating proteins; GEF: guanine-nucleotide exchange factors.

**Figure 2 biomedicines-12-01925-f002:**
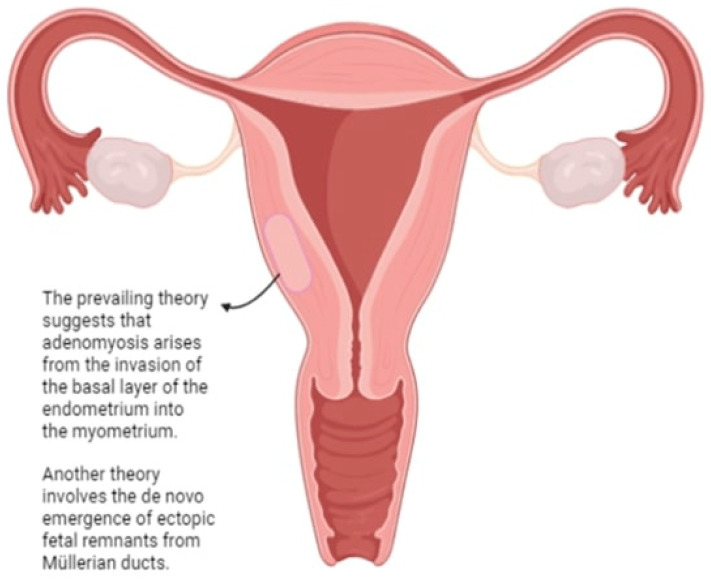
Theories explaining the origin of adenomyosis.

**Figure 3 biomedicines-12-01925-f003:**
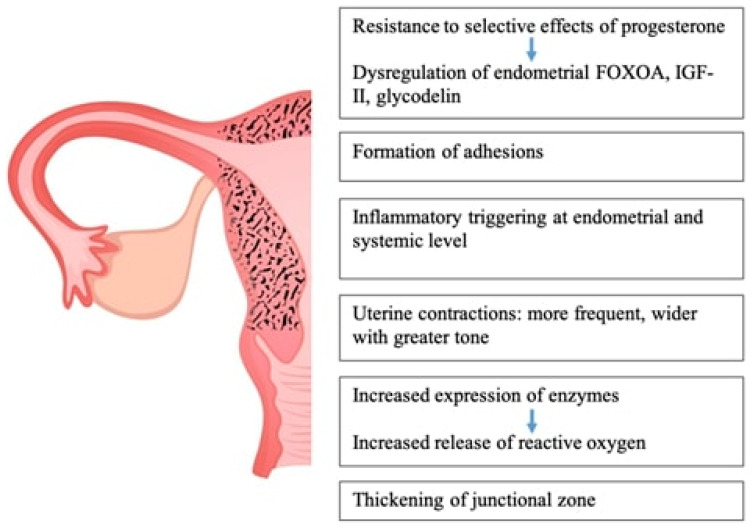
Mechanisms that redound adenomyosis in negative pregnancy outcomes.

**Table 1 biomedicines-12-01925-t001:** Complications of pregnancy associated with adenomyosis and endometriosis.

Complications	Short Notes
Preterm Birth	Failure of the normal transformation of the spiral arteries in the junction zone. Placental hypohydration seems to increase the production of pro-inflammatory mediators, resulting in local and systemic inflammation and, finally, the onset of labor [84,86].
Ectopic Pregnancy	Researchers concluded that women suffering from adenomyosis are six times more likely to experience an ectopic/tubal pregnancy [86].
SGA	Blood flow within the adenomyosis lesions is abundant, while the placenta has reduced blood flow based on the results of blood flow measurements in the myometrium and placenta of women with adenomyosis and severe SGA. Additionally, chronic inflammatory processes in the uterine microenvironment and uterine resistance to progesterone may be the possible causes of SGA [85,86,87].
Fetal Growth Restriction (FGR)	The relative risk of intrauterine growth restriction (IUGR) is 3 times higher in women with adenomyosis compared to those without the condition [86].
Hypertensive Disease of Pregnancy/Preeclampsia	Conflicting results have arisen from research studies [85].
Gestational Diabetes	Contradictory findings have emerged [86].
Placenta Previa	Women with endometriosis face a fourfold increase in the risk of developing placenta previa. It has been suggested that progesterone resistance and abnormal uterine contractility in endometriosis may contribute to delayed blastocyst implantation and embryo displacement [86].
Postpartum Hemorrhage	The heightened risk of obstetric bleeding in women with endometriosis is associated with pathological differentiation of the junctional zone, elevated concentrations of local inflammatory factors, and disordered uterine contractility. This, in turn, can lead to premature rupture of fetal membranes. Additionally, the presence of adhesions and chronic pelvic inflammation, which result in uterine fixation during deep endometriosis, further contributes to this risk [85].
Cesarean Delivery	Cesarean delivery and surgical complications, such as hysterectomy, hemoperitoneum, and bladder injuries, occur significantly more frequently in women with endometriosis [84].
Automatic Hemoperitoneum	This rare but life-threatening condition manifests either in the second trimester of pregnancy or immediately after delivery, presenting with acute or subacute abdominal pain, hemorrhagic shock, and fetal distress. It is caused either by bleeding from endometriotic foci or by the spontaneous rupture of friable tissue due to chronic inflammation, increased tension from adhesions, or enlargement of the vascular matrix [86].
Hemothorax-Pneumothorax	It is possible that the pneumothorax resulted from the rupture of adhesions between the endometriotic foci and the diaphragm [86].
Peritonitis/Bowel Perforation	Instances of intestinal rupture have been documented during the second trimester of pregnancy or shortly after childbirth. Perforations typically occur in the sigmoid colon, rectum, or appendix, although cases involving the ileum and cecum have also been observed [86].
Bladder Rupture/Uroperitoneum	Most patients with bladder endometriosis have been successfully treated without adverse effects on pregnancy outcomes. However, there is a risk of developing automatic uroperitoneum due to bladder rupture, which can result in premature labor [86].

**Table 2 biomedicines-12-01925-t002:** The postoperative fertility outcomes following surgical treatment of adenomyosis.

Authors	Conceive Rate	Conception after ART	Complications	Uterus Rupture
Wang et al., 2009 [139]	20 (74%)	-	15% miscarriage 10% preterm birth	0
Osada et al., 2011 [140]	16 (61.5%)	12 (75%)	12.5 % miscarriage	0
Sun et al., 2011 [141]	8 (33.3%)	5 (62.5%)	62.5%	0
Al Jama et al., 2011 [142]	8 (44.4%)	-	25% miscarriage	0
Saremi et al., 2014 [143]	21 (30%)	14 (66%)	19% miscarriage 6% stillbirth	2 (9%)
SHI et al., 2021 [144]	97 (53%)	70 (51.8%)	16.5% preterm birth 27.3% pregnancy loss 1.35% ectopic pregnancy 3.38% still birth 12.37% abnormal placenta	NA
Won, 2021 [145]	15 (34.9%)	14 (93.3%)	13.3% preterm birth 20% miscarriage	NA
Zhou et al., 2022 [146]	62 (45%)	27 (43.5%)	4.5% preterm birth 22.5% miscarriage	0
Ono et al., 2023 [147]	43 (95%)	26 (60.5%)	29.4% preterm birth 16.3% FGR	1 (2.3%)
Yoon et al., 2023 [148]	18 (54.5%)	NA	44.4% miscarriage 30% preterm birth	0

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
