# Peer review of "The Impact of Adenomyosis on Pregnancy"

_biomedicines, 2024, doi:10.3390/biomedicines12081925_

Round 1

Reviewer 1 Report

Comments and Suggestions for Authors

This narrative review assessed the effect of adenomyosis on pregnancy outcomes. This manuscript is informative and useful to readers. The reviewer’s comments are as follows. 

Please add the line numbers for this review. 

Figure 1: Difficult to see because of low image resolution. Please improve.

3. Pathogenesis and throughout the manuscript.

Several run-on paragraphs are excessively lengthy. Please break every to 8-12 lines.

Page 5: Figure 1: Theories explaining adenomyosis origin should be Figure 2: Theories explaining adenomyosis origin.

Table 1. Please use fetal growth restriction (FGR) instead of fetal growth retardation and intrauterine growth restriction. Please add citations for each complication.

Page 14: What is “Precursor Placenta”?

Page 14: The reviewer believes that the association between adenomyosis and placenta accreta is understudied.

Page 14: Please cite more recent studies on the definition of postpartum hemorrhage.

Page 14: Cesarean should be “Cesarean delivery.”

Page 16: The 8.3. section is incomplete.

Overall, the reviewer believes that this manuscript is informative and is worthy of publication.

Comments on the Quality of English Language

I recommend that the manuscript be reviewed by a person with professional proficiency in English to correct errors in grammar, punctuation, word choice, and sentence construction to improve the flow of ideas expressed in the article to ensure that the document reads as though written by a native English speaker.

Author Response

Dear reviewer 1,

Thank you for your recommendations. We have carefully reviewed them and implemented the following changes as a result.

  1. We added the line numbers for this review.
  2. We improved the resolution of figure 1.
  3. We have divided some of the excessively lengthy paragraphs into shorter sections.
  4. We have combined paragraphs 2 and 3 and updated the numbering throughout the text accordingly.
  5. In page 14 we meant placenta previa, so precursor placenta has changed.
  6. We believe that the analysis of the association between adenomyosis and placenta accreta is out of our manuscript’s scope.
  7. We have included two additional recent citations on postpartum hemorrhage.
  8. We changed “Cesarean” with “Cesarean delivery.”
  9. We have added the previously missing section titled "Laparoscopic Total or Subtotal Hysterectomy" and included two additional citations.

These revisions have undoubtedly improved our paper, and we're grateful for your help in making it better. Thanks again for your valuable feedback and support.

Kind regards,

Prof. Panagiotis Tsikouras

Reviewer 2 Report

Comments and Suggestions for Authors

Dear Authors,

i read your manuscript “The impact of adenomyosis on pregnancy”. The manuscript is very interesting and debating on the possible impact of disease on the pregnancy. Adenomyosis is a chronic disease with a prevalence up to 70% in pre-menopausal women. So according to this data could be important to investigate this field.

To improve the manuscript, I suggest:

1)     Abstract: no suggestions

2)     Introduction: in line 3, debating about posterior and anterior compartment and not only “such as in the area of the rectal septum “

3)     It might be useful for readers to combine paragraph 2 (origin) and paragraph 3 (pathogenesis)

4)     In the “Pathological anatomy” could be useful debating about the different form of adenomyosis (adenomyoma, focal and diffuse adenomyosis) and different degree (mild, moderate and severe) 

5)     In the “Clinical symptoms” could be useful underline the role of dysmenorrhea and possible concomitant endometriosis (this recent reference may be interesting, doi: 10.3390/jcm12175624)

6)     In the “diagnosis” could be useful underline the role of Ultrasound and MRI in the noninvasive diagnosis. it may be important to repeat the ultrasound diagnostic criteria (direct and indirect signs)

7)     In the “Adenomyosis and Pregnancy”, it could be interesting to analyze the data according to which the different forms of adenomyosis, such as internal myometrium involvement or external myiometrium involvement, have a different impact on risk of miscarriage and which major complications in pregnancy.

8)     In the “Surgical treatment of adenomyosis”, add the possible hysteroscopic treatment of the sub endometrial adenomyotic lesions and explain better the different surgical techniques to remove diffuse and focal adenomyosis, such as triple-flap method and wedge resection resection

Sincerely

Comments on the Quality of English Language

I think that a review of the language by a English native speaker could be useful to improve the manuscript

Author Response

(The authors gave the same response as above.)

Round 2

Reviewer 1 Report

Comments and Suggestions for Authors

The authors successfully addressed my previous comments. 

Comments on the Quality of English Language

The authors successfully improved the writing of the manuscript during the first round of revision.